# Active versus restrictive approach to isolated hypotension in preterm neonates: A Systematic Review, Meta-analysis and GRADE based Clinical Practice Guideline

**Viraraghavan Vadakkencherry Ramaswamy**[1◉], **Gunjana Kumar**[2◉],
**Abdul Kareem Pullattayil S**[3], **Abhishek S Aradhya**[4], **Pradeep Suryawanshi**[5],
**Mohit Sahni**[6], **Supreet Khurana**[7], **Kiran More**[8*], **on behalf of the National Neonatal Forum, India, Clinical Practice Guidelines Group on Neonatal Shock**[‡]

**1** Department of Neonatology, Ankura Hospital for Women and Children, Hyderabad, Telangana, India,
**2** Department of Neonatology, National Institute of Medical Sciences, Jaipur, Rajasthan, India, **3** Queen's University, Kingston, Canada, **4** Department of Neonatology, Ovum Women and Child Speciality Hospital, Bengaluru, Karnataka, India, **5** Department of Neonatology, Bharati Vidyapeeth University Medical College, Pune, Maharashtra, India, **6** Department of Neonatology, Surat Kids Hospital, Surat, Gujarat, India, **7** Department of Neonatology, Government Medical College and Hospital, Chandigarh, India, **8** Division of Neonatology, MRR Children's Hospital, Mumbai, Maharashtra, India

◉ These authors contributed equally to this work
‡ Collaborators of the National Neonatal Forum, India, Clinical Practice Guidelines Group on Neonatal Shock 2023 is provided in the Acknowledgements.
* drkiranmore@yahoo.com

## Abstract

### Objective

Isolated hypotension (IH) without any clinical or biochemical features of poor perfusion is a common occurrence in very preterm infants (VPTI). There exists no recommendations guiding its management. The objective of this review was to compare the effect of active vs. restrictive approach to treat IH in VPTI.

### Methodology

Medline, Embase and Web of Science were searched until 1ˢᵗ April 2024. RCTs and non-RCTs were included. Mortality, major brain injury (MBI) (intraventricular hemorrhage > grade 2 or cystic periventricular leukomalacia), mortality or neurodevelopmental impairment (NDI) at 18–24 months' corrected age were the critical outcomes evaluated.

### Results

44 studies were included: 9 were synthesized in a meta-analysis and 35 studies in the narrative review. Clinical benefit or harm could not be ruled out for the outcomes from the meta-analyses of RCTs. Meta-analysis of 3 non-RCTs suggested that active treatment of IH in VPTI of < 24 hours of life possibly increased the odds of MBI (aOR: 95% CI 1.85 (1.45; 2.36), very low certainty). Meta-analysis of 2 non-RCTs that had included VPTI < 72 hours indicated a possibly decreased risk of MBI (aOR: 95% CI 0.44 (0.24; 0.82), very

**Data availability statement:** All relevant data are within the manuscript and its Supporting Information files.

**Funding:** The author(s) received no specific funding for this work.

**Competing interests:** The authors have declared that no competing interests exist.

low certainty) and NEC ≥ stage 2 (aOR: 95% CI 0.61 (0.41; 0.92), very low certainty) with active treatment of IH. Active treatment of IH in the first 24 hours possibly increased the risk of mortality or long-term NDI (aOR: 95% CI 1.84 (1.10; 3.09), very low certainty) and the risk of hearing loss at 2 years (aOR: 95% CI 3.60 (1.30; 9.70), very low certainty). Clinical benefit or harm could not be ruled out for other outcomes. There was insufficient evidence with respect to preterm neonates of ≥ 32 weeks.

## Conclusions

IH may not be treated in VPTI in the first 24 hours. However, IH occurring between 24 hours - 72 hours of life may be treated. The evidence certainty was very low.

## Introduction

Isolated hypotension (IH) without any clinical or biochemical features of poor perfusion is a common occurrence in very preterm infants (VPTI) and extremely low gestational age neonates (ELGANs) in the initial days of life. [1,2] The definition of hypotension based on blood pressure (BP) values is a controversial topic. [3] The systematic review by Dempsey et al. concluded that that there was paucity of data as to whether any intervention for hypotensive neonates translated to improved outcomes. [4] Two randomised controlled trials (RCTs) published subsequently compared active treatment (IH without any clinical or biochemical signs of poor perfusion) vs. restrictive treatment (no treatment or treatment only when clinical signs of poor perfusion was present) in VPTI and ELGANs. [1,2] Both the trials were underpowered, and one of them was stopped prematurely. [1] Further, a pilot trial by Batton et al. pointed out the various bottlenecks associated with conducting such trials, namely the low consent rates and lack of physician equi-poise. [5] A scoping review of literature indicated that many observational studies have been published on this topic since the last systematic review by Dempsey et al. [6–10] The results of these studies were contentious with widely differing conclusions. The Grading of Recommendations, Assessment, Development and Evaluations (GRADE) working group had indicated that in such scenarios though not making any recommendations may be an option, it emphasized that such an approach is not advisable. [11] The argument for the same was that clinicians would rarely search for evidence systematically as thoroughly as a guideline panel, neither would they have the time or resources to evaluate the possible underlying values and preferences of the parents. [12] Till date, there exists no clinical practice guideline (CPG) that had addressed this topic of management of IH in preterm neonates. Thus, it is imperative that the published literature on this contentious topic be evaluated through a validated process to guide safe clinical practice, with an aim to improve neonatal outcomes. Henceforth, this systematic review and meta-analysis was performed. A CPG was formulated following the stringent GRADE working group guidelines. [13]

## Methods

The protocol was registered with PROSPERO (https://www.crd.york.ac.uk/prospero/display_record.php?ID=CRD42023446821) and the reporting of this systematic review adheres to the PRISMA guidance.

### Inclusion criteria

**Population(P).** Preterm neonates of less than 37 weeks' gestational age within the first week of life with IH. IH was defined as low mean arterial blood pressure (MAP) as ascertained by the investigator which could be based on different definitions such as MAP less than

gestational age, MAP value below a particular centile for the gestational age, MAP less than 30 mm Hg, and without any clinical or biochemical evidence of hypoperfusion.

**Intervention(I) (active treatment group).** Treatment with inotropes for IH. Volume expansion with crystalloids or colloids could have been used prior to inotrope or vasopressor initiation.

**Comparator(C) (restrictive treatment group).**

a. Treatment with volume expansion and/ or inotropes in preterm neonates with hypotension only when clinical and/ or biochemical features of poor perfusion were present. The clinical and/ or biochemical signs of hypoperfusion were defined as presence of either of these: Unexplained tachycardia (>160–170 beats/min)[14], prolonged capillary refilling time (> 3–4 seconds)[15,16], low peripheral pulses, decreased urine output (< 1 ml/kg/ hour for 4 -6 hours, physiological oliguria or anuria should also be considered) [17], increasing lactate levels (> 3–4 mmol/L)[15] and base deficit (> 8 meq/L). [2,7]

b. No treatment of IH.

**Outcomes(O).**

a. **Critical outcomes:** Mortality, major brain injury (MBI) (defined as intraventricular hemorrhage (IVH)> stage 2 and/ cystic periventricular leukomalacia (PVL)), composite outcome of mortality or neurodevelopmental impairment (NDI) at 18–24 months' corrected age (CA).

b. **Important outcomes:** Bronchopulmonary dysplasia (BPD) (respiratory support requirement at 36 weeks' postmenstrual age (PMA)), necrotising enterocolitis (NEC) ≥ stage 2, patent ductus arteriosus (PDA) requiring treatment and retinopathy of Prematurity (ROP) requiring treatment.

**Study designs:** RCTs and observational studies were eligible for inclusion. Case reports, descriptive reviews and case series were excluded.

**Time frame:** From inception of the databases until 1$^{st}$ April 2024

**Literature search strategy.** Medline, Embase and Web of Science were searched from inception until 1$^{st}$ April 2024. (S1a Table, S1b Table in S1 File) After the removal of duplicates, titles and abstracts were screened for potentially eligible studies. Full texts of the respective studies were assessed for possible inclusion. Two authors blinded to each other performed the literature search. Disagreements were resolved by consensus. Reference lists of included studies and other similar systematic reviews were hand searched for studies that satisfied the inclusion criteria. There were no language restrictions. Only published literature was included.

**Data extraction and synthesis.** Two authors extracted data independently. Data synthesis was performed using the R-software (version 2023.06.0 + 421) (R Foundation for Statistical Computing, Vienna, Austria). [18] A random-effects model with weighted average (Mantel-Haenszel method) for pooling raw data from RCTs and inverse variance method for pooling of adjusted odds ratio (aOR) with 95% confidence interval (CI) was utilized. Raw unadjusted data from observational studies were included in the narrative review. Statistical heterogeneity was assessed based on I² test, Tau² (using the DerSimonian-Laird estimator) and Cochran Q. For any outcome, if ten or more studies were available for meta-analysis, publication bias was planned to be assessed. In cases of missing data for a particular outcome, we decided to adopt the strategy of last observation carry forward (LOCF).

**Risk of bias.** Risk of bias was performed using the Cochrane Risk of Bias tool version 2.0 for RCTs [19] and Risk Of Bias in Non-randomized Studies-of Intervention (ROBINS-I)

for non-RCTs [20] by two authors blinded to each other. Disagreements were resolved by consensus.

**Certainty of evidence and recommendations.** Certainty of evidence (CoE) for the effect estimates of outcomes was assessed according to GRADE. [13] The reporting of the results of the systematic review was done as per the modified GRADE recommendations. [21] (**S2 Table** in S1 File) Evidence to decision framework (EtD) was used to arrive at the recommendations. [11]

## Results

After the removal of duplicates, 2166 titles and abstracts were screened. Of these, a total of 44 studies were included in the systematic review: 35 studies [3,5,6,20,22–52] were included in the narrative review (RCT:1 [5], observational studies: 34) and 9 in meta-analysis (RCTs: 2 [1,2], observational studies: 7 [7–10,53–55]). The PRISMA flow is depicted in **Fig 1**.

Amongst the 35 studies in included in the narrative review, 12 studies had included ELGANs [3,5,6,23–25,30,36,47,50,52,56], 16 studies VPTI [22,26,27,29,33–35,37–41,43,48,49,51], and 7 studies had evaluated preterm neonates of > 32 weeks' gestation [28,31,32,42,44–46]. Of the 9 studies included in the meta-analysis, 5 studies had enrolled ELGANs [1,7,9,53,54] and 4 VPTI [2,8,10,55]. The definition used for hypotension varied between the studies. The commonly used definitions were mean arterial pressure (MAP) < gestational age in weeks, MAP < 30 mm Hg, MAP < 25 mm Hg and MAP less than $3^{rd}$ –$10^{th}$ centile for gestational age, birth weight and postnatal age, and MAP < median MAP based on centiles according to the gestational age and the postnatal age. Some studies had defined hypotension based on clinical or biochemical signs

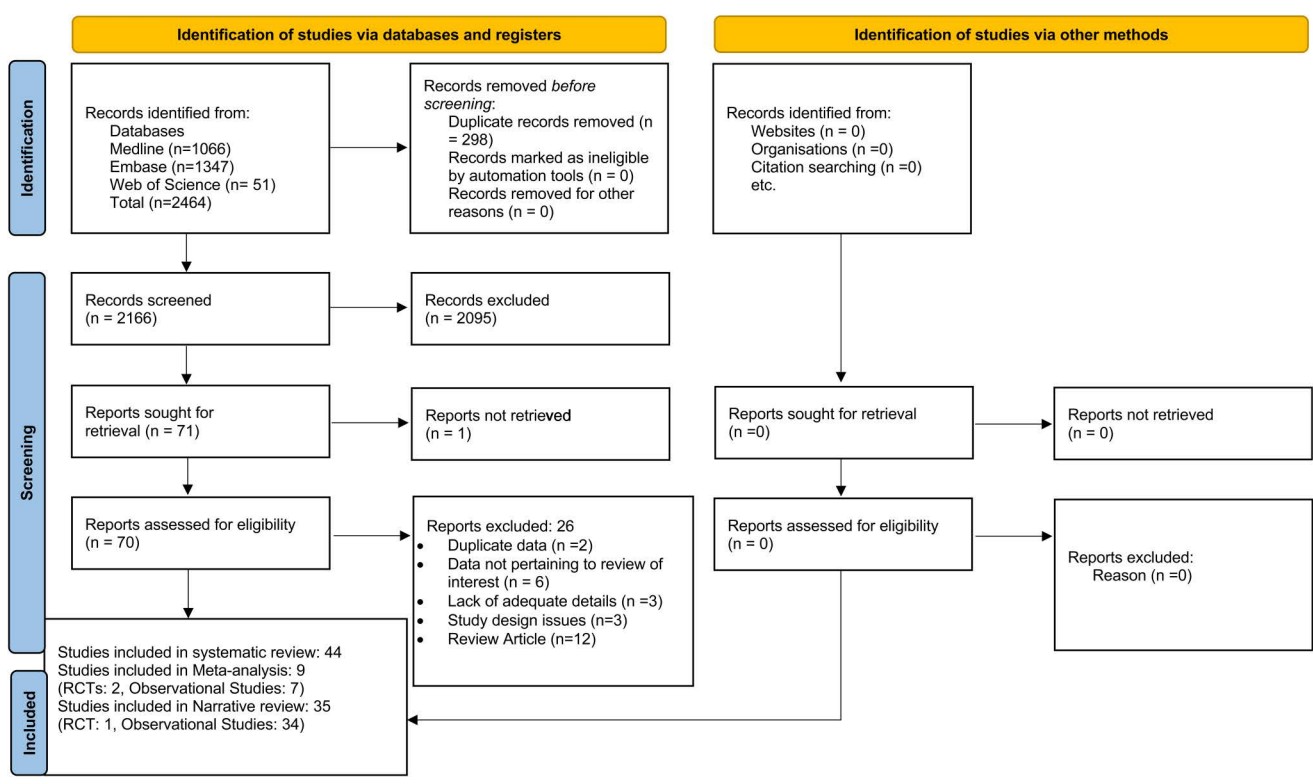

**Fig 1. PRISMA flow.**

of poor perfusion with or without a MAP cut-off. [28,30,33] The characteristics of the included studies is provided in S1 Table and S2 Table in S2 File.

### Risk of bias

Of the three included RCTs [1,5,53], 2 had a high risk of bias [1,2] and one had a low risk of bias. [5] The RCTs with a high risk of overall bias had issues with the domain of deviation from the intended interventions. Amongst the observational studies, most of the studies had a moderate to serious risk of bias due to issues with the domains of confounding and classification of interventions. (S3 Table in S1 File)

### Outcomes

**Mortality.** Clinical benefit or harm could not be ruled out for the outcome of mortality from meta-analysis of RCTs [Risk ratio (RR): 1.08, 95% confidence interval (CI): 0.48–2.43; 2 RCTs; n = 118] [1,2] or non RCTs [Adjusted odds ratio (aOR): 0.71, 95% CI: 0.40–1.03; 3 non-RCTs; n = 5816] [7–9] due to statistically non-significant results and the CoE being very low to low. (Fig 2, Table 1)

**Major brain injury.** Clinical benefit or harm could not be ruled out for the outcome of MBI from meta-analysis of RCTs (RR: 1.00, 95% CI: 0.32–3.09; 2 RCTs; n = 118).[1,2] When aORs with 95% CI from non-RCTs were pooled for the outcome of MBI, significant heterogeneity was detected ($I^2$: 88%). Sub-group analysis based on the postnatal age of the neonates possibly explained the between studies heterogeneity. While active treatment of IH in the first 24 hours possibly increased the odds of MBI (aOR: 1.85, 95% CI: 1.45 - 2.36; 3 non-RCTs; n = 5495) (8,54,55), meta-analysis of observational studies which had included neonates of < 72 hours of life indicated that active treatment of IH possibly decreased the odds of MBI (aOR:0.44, 95% CI: 0.24 - 0.82, 2 non-RCTs, n = 909). [7,9] (Fig 2)

**NEC ≥ stage 2..** Clinical benefit of harm could not be ruled out for this outcome from the meta-analysis of RCTs (RR: 0.67, 95% CI: 0.28–1.59; 2 RCTs; n = 118). Meta-analysis of the observational studies which had enrolled neonates of < 72 hours indicated that active treatment of IH possibly decreased the risk of NEC ≥ stage 2 (aOR: 0.61; 95% CI: 0.41 - 0.92, 2 non-RCTs, n = 909). [7,9] (Fig 2, Table 1)

**BPD.** There was a trend towards a possibly increased risk of BPD with active approach to management of IH from the meta-analysis of RCTs (RR:1.19, 95% CI: 0.86 - 1.64; 2 RCTs; n = 118) [1,2] and that of non-RCTs (aOR: 1.29, 95% CI: 0.98 - 1.71; 4 non-RCTs; n = 5805) [7–9,55]. (Fig 2, Table 1)

**ROP requiring treatment.** Clinical benefit or harm could not be ruled out from the only non-RCT that had reported on this outcome (aOR: 0.46, 95% CI: 0.04–5.15; n = 191) [9]

**PDA requiring treatment (medical or surgical).** Clinical benefit or harm could not be ruled out for this outcome from meta-analysis of RCTs (RR:1.12, 95% CI: 0.59–2.15; 2 RCTs; n = 118). A single non-RCT enrolling 671 neonates indicated that the active management of IH possibly increased the odds of PDA requiring treatment (aOR: 1.64, 95% CI: 1.12 - 2.44). [7] (Fig 2, Table 1)

**Mortality or neurodevelopmental impairment at 18–22 months' CA.** Very low evidence certainty from one non-RCT enrolling 137 neonates suggested that active treatment of IH in the first 24 hours might possibly increase the risk of mortality or NDI (aOR: 1.84, 95% CI: 1.10 - 3.09). [53] One non-RCT with a case control design enrolling 25 cases and 710 controls indicated that an active approach to management of IH in the first 24 hours of life was associated with an increased risk of sensineural hearing loss at 2 years (aOR: 3.60, 95% CI: 1.30 - 9.70, very low certainty)[55] (Table 1)

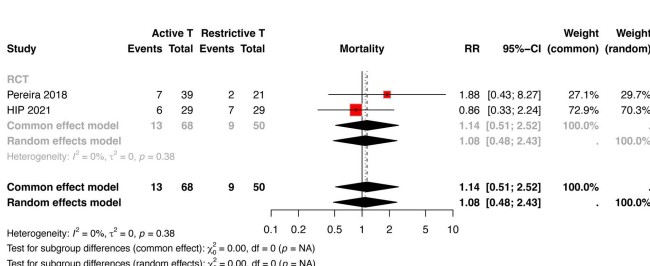

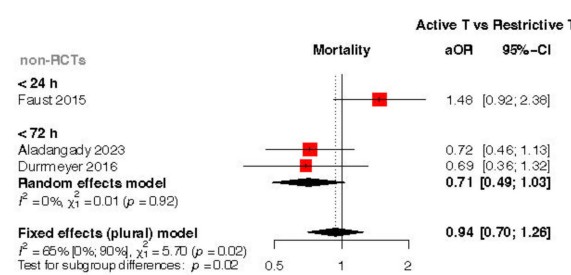

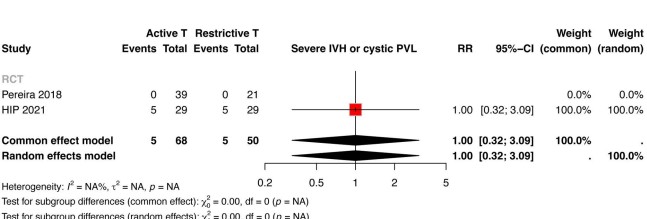

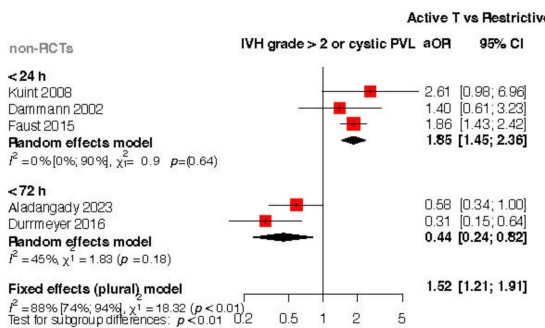

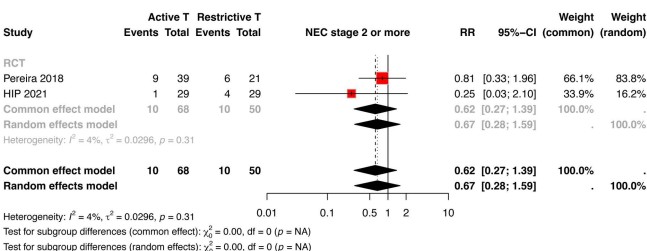

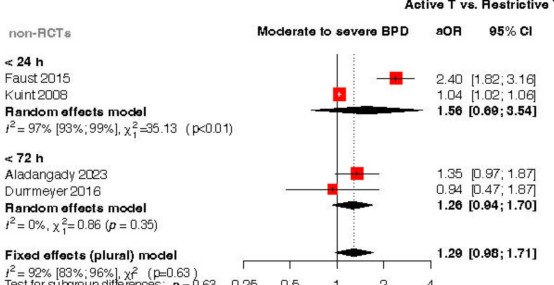

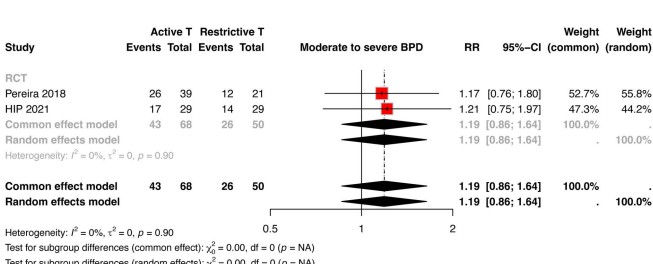

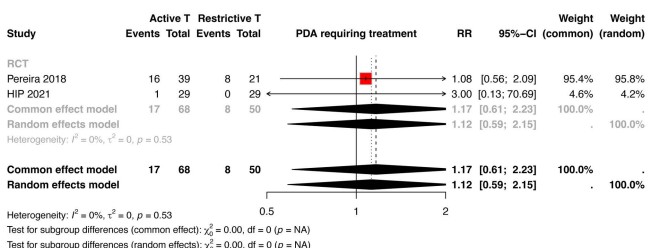

RR: Risk ratio
95%-CI: 95% confidence interval
T: Treatment
h: hours
RCTs: Randomized controlled trials
aOR: Adjusted odds ratio
IVH: Intraventricular hemorrhage
NEC: Necrotising enterocolitis
BPD: Bronchopulmonary dysplasia
PDA: Patent ductus arteriosus
NA: Not applicable

**Fig 2. Forest plot depicting the effect estimates from randomized controlled trials and non-randomized controlled trials for the various outcomes.**

**Table 1.** Active treatment (isolated hypotension without any clinical or biochemical signs of poor perfusion) compared to restrictive treatment (no treatment of isolated hypotension or treatment of hypotension with clinical or biochemical signs of poor perfusion) in preterm neonates born at less than 37 weeks' gestation in the first week of postnatal life.

| Certainty assessment | | | | | | | Summary of findings | | | | |
|---|---|---|---|---|---|---|---|---|---|---|---|
| Participants (studies) Follow-up | Risk of bias | Inconsistency | Indirectness | Imprecision* | Publication bias | Overall certainty of evidence | Study event rates (%) | | Relative effect (95% CI) | Anticipated absolute effects | |
| | | | | | | | With restrictive treatment (no treatment of isolated hypotension or treatment of hypotension with clinical or biochemical signs of poor perfusion) | With active treatment (isolated hypotension without any clinical or biochemical signs of poor perfusion) | | Risk with restrictive treatment (no treatment of isolated hypotension or treatment of hypotension with clinical or biochemical signs of poor perfusion) | Risk difference with active treatment (isolated hypotension without any clinical or biochemical signs of poor perfusion) |
| **Neonatal mortality (RCTs)** | | | | | | | | | | | |
| 118 (2 RCTs) [1,2] | serious[a] | not serious | serious[b] | very serious[c] | none | ⊕◯◯◯ Very low | 9/50 (18%) | 13/68 (19.1%) | **RR 1.08** (0.48 to 2.43) | 180 per 1,000 | 14 more per 1,000 (from 94 fewer to 257 more) |
| **Neonatal mortality (non-RCTs)** | | | | | | | | | | | |
| 5816 (3 observational studies)[9,24,54] | serious[d] | serious[e] | serious[b] | very serious[f] | none | ⊕◯◯◯ Very low | 220/2815 (7.8%) | 207/3001 (6.9%) | **OR 0.71** (0.49 to 1.03) | 78 per 1,000 | 21 fewer per 1,000 (from 38 fewer to 2 more) |
| **Severe brain injury (defined as IVH > grade 2 or cystic PVL) (RCTs)** | | | | | | | | | | | |
| 118 (2 RCTs)[1,2] | serious[a] | not serious | serious[b] | extremely serious[g] | none | ⊕◯◯◯ Very low | 5/50 (10.0%) | 5/68 (7.4%) | RR 1.00 (0.32 to 3.09) | 100 per 1,000 | 0 fewer per 1,000 (from 68 fewer to 209 more) |
| **Severe brain injury (defined as IVH > grade 2 or cystic PVL) (non-RCTs)** | | | | | | | | | | | |
| 5993 (5 observational studies) [9,24,54,55,54] | serious[h] | very serious[i] | serious[b] | very serious[j] | none | ⊕◯◯◯ Very low | 544/2909 (18.7%) | 459/3084 (14.9%) | **OR 1.52** (1.21 to 1.91) | 187 per 1,000 | 72 more per 1,000 (from 31 more to 118 more) |
| **NEC ≥ stage 2 (RCTs)** | | | | | | | | | | | |
| 118 (2 RCTs) [1,2] | not serious | not serious | serious[b] | extremely serious[g] | none | ⊕◯◯◯ Very low | 10/50 (20.0%) | 10/68 (14.7%) | RR 0.67 (0.28 to 1.59) | 200 per 1,000 | 66 fewer per 1,000 (from 144 fewer to 118 more) |
| **NEC ≥ stage 2 (non-RCTs)** | | | | | | | | | | | |
| 908 (2 observational studies) [9,54] | serious[k] | not serious | serious[b] | very serious[j] | none | ⊕◯◯◯ Very low | 91/526 (17.3%) | 41/382 (10.7%) | **OR 0.61** (0.41 to 0.92) | 173 per 1,000 | 60 fewer per 1,000 (from 94 fewer to 12 fewer) |
| **PDA requiring treatment (RCTs)** | | | | | | | | | | | |
| 118 (2 RCTs) [1,2] | not serious | not serious | serious[b] | extremely serious[g] | none | ⊕◯◯◯ Very low | 8/50 (16.0%) | 17/68 (25.0%) | RR 1.12 (0.59 to 2.15) | 160 per 1,000 | 19 more per 1,000 (from 66 fewer to 184 more) |
| **PDA requiring treatment (non-RCT)** | | | | | | | | | | | |
| 671 (1 observational study) [54] | very serious[l] | not serious | serious[b] | very serious[j] | none | ⊕◯◯◯ Very low | 64/408 (15.7%) | 63/263 (24.0%) | **OR 1.64** (1.12 to 2.44) | 157 per 1,000 | 77 more per 1,000 (from 16 more to 155 more) |
| **Moderate to severe BPD (O2 or respiratory support requirement at 36 weeks' PMA) (RCTs)** | | | | | | | | | | | |
| 118 (2 RCTs)[1,2] | not serious | not serious | serious[b] | very serious[c] | none | ⊕◯◯◯ Very low | 26/50 (52.0%) | 43/68 (63.2%) | RR 1.19 (0.86 to 1.64) | 520 per 1,000 | 99 more per 1,000 (from 73 fewer to 333 more) |

*(Continued)*

**Table 1.** (Continued)

| Certainty assessment | | | | | | | Summary of findings | | | | |
|---|---|---|---|---|---|---|---|---|---|---|---|
| Participants (studies) Follow-up | Risk of bias | Inconsistency | Indirectness | Imprecision* | Publication bias | Overall certainty of evidence | Study event rates (%) | | Relative effect (95% CI) | Anticipated absolute effects | |
| | | | | | | | With restrictive treatment (no treatment of isolated hypotension or treatment of hypotension with clinical or biochemical signs of poor perfusion) | With active treatment (isolated hypotension without any clinical or biochemical signs of poor perfusion) | | Risk with restrictive treatment (no treatment of isolated hypotension or treatment of hypotension with clinical or biochemical signs of poor perfusion) | Risk difference with active treatment (isolated hypotension without any clinical or biochemical signs of poor perfusion) |
| **Moderate to severe BPD (O2 or respiratory support requirement at 36 weeks' PMA (non-RCTs)** | | | | | | | | | | | |
| 5805 (4 observational studies) [9,24,54,55] | serious[k] | very serious[h] | serious[b] | very serious[c] | none | ⊕◯◯◯ Very low | 601/2715 (22.1%) | 562/3090 (18.2%) | **OR 1.29** (0.98 to 1.71) | 221 per 1,000 | 47 more per 1,000 (from 3 fewer to 106 more) |
| **Severe ROP (non-RCT)** | | | | | | | | | | | |
| 191 (1 observational study) [9] | serious[l] | not serious | Serious[b] | very serious[m] | none | ⊕◯◯◯ Very low | 2/92 (2.2%) | 1/99 (1.0%) | **RR 0.46** (0.04 to 5.15) | 22 per 1,000 | 12 fewer per 1,000 (from 21 fewer to 90 more) |
| **Mortality or neurodevelopmental impairment at 18–22 months' corrected age (non-RCT)** | | | | | | | | | | | |
| 137 (1 observational study) [24] | serious[l] | not serious | Serious[b] | serious[j] | none | ⊕◯◯◯ very low | 19/67 (28.4%) | 33/70 (47.1%) | **OR 1.84** (1.10 to 3.09) | 284 per 1,000 | 138 more per 1,000 (from 20 more to 267 more) |
| **Sensineural hearing loss at 12–24 months' corrected age (non-RCT)** | | | | | | | | | | | |
| 25 cases 710 controls (1 observational study) [55] | very serious[j] | not serious | serious[b] | serious[j] | none | ⊕◯◯◯ Very low | 25 cases 710 controls | | **OR 3.60** \(1.30 to 9.70) | - | |

Explanations

[a] One study which had the highest weightage had a high risk of bias.

[b] Indirectness related to patient population (definition of the outcome hypotension) and additional interventions in either of the arms.

[c] The minimally important difference (MID) indicates probable important harm, but the boundary of CI suggests possible important benefit as well.

[d] While 2 studies had a serious risk of overall bias, one had a moderate risk of overall bias.

[e] $I^2 > 50\%$

[f] The MID indicates probable important benefit, but the boundary of CI suggests possible harm as well.

[g] The CI boundary is very wide and is uncertain.

[h] Of the 5 studies, 2 had moderate risk of overall bias and 3 had serious risk of overall bias.

[i] $I^2 > 75\%$.

[j] The ratio of OR 95%CI boundary is >1.5

[k] While one study had a serious risk of overall bias, the other had a moderate risk of overall bias.

[l] The single study had a serious risk of overall bias.

[m] Trivial effect, but boundaries of CI indicate both probable important harm and possible important benefit.

* MID set for critical outcomes: 1%, important outcomes: 5%

The narrative review of the studies included in the systematic review is provided in **S1_file Appendix** in S1 File. A table of all data extracted from the primary studies included in this systematic review and meta-analyses is provided in S5 Table in S1 File.

### Evidence to decision

The EtD framework using the GRADE guidelines was used to arrive at recommendations. The parameters of desirable effects, undesirable effects, evidence certainty, values of the outcomes

studied, balance of effects, resources required, cost-effectiveness, equity, acceptability, and feasibility were taken into consideration. These are detailed in S4 Table in S1 File.

## Discussion

In this systematic review and meta-analysis, we evaluated 44 studies, predominantly including ELGANs and VPTI in the first 72 hours of life with hypotension. Two interventions, namely active treatment vs. restrictive treatment of IH were compared. The results of this systematic review and meta-analysis were evaluated through the GRADE framework to make recommendations and formulate a CPG. To the best of our knowledge, there exists no CPG for this specific PICO.

It should be noted that there was some clinical heterogeneity with regards to the patient population, the intervention and the comparator in the included studies in the meta-analyses for the various outcomes. The question of whether pooling of data in a meta-analysis arises in such scenarios. The GRADE working group provides clarity with respect to that. While GRADE visualizes the variability between studies as a source of potential opportunity. GRADE group points out that in some scenarios such variability might even turn out to be one of the strengths of a systematic review.[57] In our systematic review, the meta-analyses of RCTs did not show any between studies heterogeneity though there were some differences in the patient population, intervention and the comparator. This implies that the results of the meta-analysis of the RCTs could be generalizable to different clinical settings where isolated hypotension is defined based various definitions, the intervention and the comparator could deviate slightly still yielding similar clinical outcomes. Further, the meta-analyses of non-RCTs indicated significant heterogeneity for the critical outcome of mortality and the important outcome of severe grade IVH. This provided us an opportunity to explore the various potential reasons for the same. We found postnatal age to be an important effect modifier and hence we could bring out recommendations based on postnatal age of the neonate (≤24 h vs. > 24 h). Such pragmatic systematic reviews form the basis for formulation of clinical practice guidelines for difficult questions that often addressed through expert consensus. Further, these systematic reviews could be generalizable across different settings and also provide clinicians, parents and other stakeholders to choose between different treatment choices.

Based on the aforementioned parameters, the guideline development group suggests that clinicians may not treat IH (hypotension without any clinical and/ or biochemical signs of poor perfusion) in VPTI in the first 24 hours of life, the evidence certainty being very low and the recommendation being weak.

Further, our consensus was that the MAP cut-off to define hypotension may be centile-based. The gestational age-based centiles derived from the largest cohort of VPTI from the German Neonatal Network may be used in the first 24 hours of life.[8] To provide a safety net, we suggest that severe IH with a MAP value of less than 5 mm Hg (median MAP – 5 mm Hg) for the corresponding gestational age may be treated in VPTI in the first 24 hours of life.

Further, we suggest that in VPTI, IH beyond the first 24 hours of life and within the first 72 hours may be treated. The MAP cut-off for treatment of IH beyond 24 hours of life may be a MAP of less than the neonates' gestational age in weeks. The evidence certainty was very low, and the strength of the recommendation was weak. There was insufficient evidence to recommend for or against the treatment of IH in preterm neonates of ≥ 32 weeks' gestation.

The justification for these recommendations were based on careful consideration of all aspects of the EtD framework. Active treatment of IH in VPTI in the first 24 hours of life was shown to be associated with an increased risk of MBI, mortality or NDI at 18–22

months' CA and hearing loss at 2 years' CA. Studies included in the narrative review had also reported similar findings. Further, studies have shown that in VPTI, the MAP shows a slow physiological rise over the first day.[23,53] Actively intervening with an aim to increase the MAP might result increased MAP variability which has also been shown to be associated with poorer short- and long-term outcomes in many of the included studies. [22,27,28] To substantiate this argument, some of the included studies on cerebral blood flow (CBF) measurement in VPTI of < 24 hours of life have shown that CBF is affected only at a significantly lower MAP, and also that a higher MAP during this transitional period may be counterproductive as some studies have shown compromised cerebral perfusion at relatively higher MAP. [38,43] Since sub-group analyses indicated that active treatment of IH in VPTI between 24 hours to 72 hours of age was associated with a decreased risk of NEC ≥ stage 2 and MBI, separate recommendations were made based on the postnatal age.

The largest cohort study (German Neonatal Network) has published nomograms of median mean arterial blood pressures for VPTI in the first 24 hours of life. [8] In this study, there was a strong association between not treating severe IH (when the cut-off MAP used was median MAP for the corresponding GA minus 5 mm Hg) and poor short-term outcomes. This cut-off was also used by the HIP trial in the placebo group as rescue therapy.[1] Hence, this MAP cut-off was chosen for rescue therapy for neonates with severe IH in the first 24 hours of life.

The implementation of these recommendations needs the following good practice statements to be considered. The studies included in the meta-analysis have used both invasive and non-invasive methods for BP monitoring. If feasible invasive BP monitoring may be preferred.[58] Further, preterm neonates with IH who are treated by a restrictive approach should be closely monitored for clinical and/ or biochemical signs of hypoperfusion such as unexplained persistent tachycardia [14], prolonged capillary refilling time [15,16], low peripheral pulses, decreased urine output [17], increasing lactate levels [15] and base deficit [2,7]. If feasible, functional echocardiography may be utilized as an adjunct to the clinical and biochemical criteria.[59] Warm shock may mimic isolated hypotension, and hence those neonates at risk of early onset sepsis should be closely monitored.[60] If the decision to treat IH is made based on clinicians' discretion, the target MAP should be just more than or equal to the neonates' gestational age in weeks as higher MAP and increased variability in blood pressures have been shown to be associated with poor CBF potentially resulting in MBI. There were several limitations to this meta-analysis. The definition of IH and clinical hypotension with signs of hypoperfusion varied between studies. Further, some of the neonates in the restrictive approach group received volume expansion which could have altered the effect estimates of the meta-analyses. We tried to address these by downrating the evidence by one level for the domain of indirectness related to patient population and the intervention/ comparator.

## Conclusions

In conclusion, the recommendations of the guideline development group were based on very low evidence certainty. Large multi-center RCTs comparing active versus restrictive treatment of IH in neonates of different gestational age cohorts at varying post-natal ages are needed. Low consent rates and lack of physician equipoise are the major barriers for conducting such trials.[5] Though it has been suggested that such a trial may satisfy all the criteria for waiver of consent, this aspect should be thoroughly evaluated by the Ethical Committee of the respective centers.

## Supporting information

**S1 File.** **S1 Table**. Literature search strategy. **S2 Table**. Modified GRADE approach for reporting of results of the systematic review. **S3 Table**. Risk of bias assessment of the included studies. **S4 Table.** Evidence to Decision framework. **Appendix** Narrative review of included studies. **S5 Table**: Data extraction table.
(PDF)

**S2 File.** **S1 Table**. Characteristics of the studies included in the meta-analysis. **S2 Table**. Characteristics of the studies included in the narrative review.
(DOCX)

**S1 Checklist.** **PRISMA_2020_checklist-2.**
(PDF)

## Acknowledgement

Drs VVR, GK, AAS, PS, MS, SK, KM and the National Neonatal Forum (NNF), India, Clinical Practice Guidelines (CPG) Group 2023, conceptualized this systematic review, meta-analysis and CPG. Mr. AKPS formulated the literature search strategy and was involved in the data curation. Drs GK and VVR did the literature search, data curation and data analysis. Dr VVR produced the initial draft. Drs AAS, PS, MS, SK and KM provided further intellectual inputs and revised the initial draft. All the authors approved the final version of the manuscript submitted for peer review.

Collaborators of the NNF, India, CPG Group on Neonatal Shock 2023: We acknowledge the valuable contributions of Shiv Sajan Saini[1], Ravishankar K[2], Shashi Kant Dhir[3], Deepak Chawla[4] and Praveen Kumar[1] who collaborated as Reviewers and Editorial Coordinators in the formulation of these recommendations based on the GRADE framework. Deepak Sharma[5] collaborated in the conceptualization of this CPG.

The lead author of the NNF, India, CPG Group 2023 was Praveen Kumar, phone no.: +919478366925, e-mail: drpkumarpgi@gmail.com.

This work was undertaken as part of the development of CPGs by NNF, India. The executive summary of the final recommendations is available in the NNFI website. All the data used in this systematic review were from published literature and can be shared on reasonable request.

Affiliations of Collaborators:

[1]Department of Neonatology, Postgraduate Institute of Medical Education and Research, Chandigarh, India

[2]Department of Neonatology, Sowmya Children's Hospital, Hyderabad, Telangana, India.

[3]Department of Pediatrics, Guru Gobind Medical College, Faridkot, Punjab, India.

[4]Department of Neonatology, Government Medical College and Hospital, Chandigarh, India

[5] Department of Neonatology, National Institute of Medical Sciences, Jaipur, Rajasthan, India

## Author contributions

**Conceptualization:** Viraraghavan Vadakkencherry Ramaswamy, Gunjana Kumar, Abdul Kareem Pullattayil S, Abhishek S Aradhya, Pradeep Suryawanshi, Mohit Sahni, Supreet Khurana, Kiran More.

**Data curation:** Viraraghavan Vadakkencherry Ramaswamy, Gunjana Kumar, Abdul Kareem Pullattayil S, Supreet Khurana, Kiran More.

**Formal analysis:** Viraraghavan Vadakkencherry Ramaswamy, Gunjana Kumar, Abhishek S Aradhya, Pradeep Suryawanshi, Mohit Sahni, Supreet Khurana, Kiran More.

**Investigation:** Viraraghavan Vadakkencherry Ramaswamy, Gunjana Kumar, Pradeep Suryawanshi, Kiran More.

**Methodology:** Viraraghavan Vadakkencherry Ramaswamy, Gunjana Kumar, Abdul Kareem Pullattayil S, Pradeep Suryawanshi, Mohit Sahni, Supreet Khurana, Kiran More.

**Project administration:** Abhishek S Aradhya.

**Resources:** Abdul Kareem Pullattayil S, Abhishek S Aradhya.

**Software:** Gunjana Kumar, Abdul Kareem Pullattayil S, Abhishek S Aradhya.

**Supervision:** Pradeep Suryawanshi, Mohit Sahni, Supreet Khurana, Kiran More.

**Validation:** Abdul Kareem Pullattayil S, Kiran More.

**Writing – original draft:** Viraraghavan Vadakkencherry Ramaswamy, Supreet Khurana.

**Writing – review & editing:** Gunjana Kumar, Abdul Kareem Pullattayil S, Abhishek S Aradhya, Pradeep Suryawanshi, Mohit Sahni, Supreet Khurana, Kiran More.

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
