## [Decision Letter · Decision Letter 0]

1 Apr 2024

PONE-D-23-44090Active Versus Restrictive Approach To Isolated Hypotension In Preterm Neonates: A Systematic Review And Meta-analysisPLOS ONE

Dear Dr. More,

Thank you for submitting your manuscript to PLOS ONE. After careful consideration, we feel that it has merit but does not fully meet PLOS ONE’s publication criteria as it currently stands. Therefore, we invite you to submit a revised version of the manuscript that addresses the points raised during the review process.

We look forward to receiving your revised manuscript.

Kind regards,

Sanjoy Kumer Dey, M.D

Academic Editor

PLOS ONE

2. Please amend either the title on the online submission form (via Edit Submission) or the title in the manuscript so that they are identical.

3. Please amend the manuscript submission data (via Edit Submission) to include author Abdul Kareem Pullattayil S.

4. One of the noted authors is a group [National Neonatal Forum (NNF), Clinical Practice Guidelines (CPG) Group, India]. In addition to naming the author group, please list the individual authors and affiliations within this group in the acknowledgments section of your manuscript. Please also indicate clearly a lead author for this group along with a contact email address.

5. Please include your tables as part of your main manuscript and remove the individual files. Please note that supplementary tables (should remain/ be uploaded) as separate ""supporting information"" files". 

Reviewers' comments:

Reviewer's Responses to Questions

**Comments to the Author**

1. Is the manuscript technically sound, and do the data support the conclusions?

Reviewer #1: Yes

Reviewer #2: No

2. Has the statistical analysis been performed appropriately and rigorously? 

Reviewer #1: Yes

Reviewer #2: No

3. Have the authors made all data underlying the findings in their manuscript fully available?

Reviewer #1: Yes

Reviewer #2: Yes

4. Is the manuscript presented in an intelligible fashion and written in standard English?

Reviewer #1: Yes

Reviewer #2: Yes

5. Review Comments to the Author

Reviewer #1: THis is good review on the topic Active versus Restrictive managment of Isolated Hypotension in preterm infants. Although the evidence evalauted did not result in strong recommendations, the authors suggest the need for well conducted RCT on this important topic.

I feel the paragraphs on the narrative review may be deleted altogether to improve the readablilty of the study. However some sections of the narrative review may be used by the authors in the discussion to understand how some the weak or expert recommendations were arrived at.

Reviewer #2: Active Versus Restrictive Approach to Isolated Hypotension In Preterm Neonates: A Systematic Review And Meta-analysis

The authors tried to address an important yet les studied and controversial topic. They have systematically approached the topic and must be congratulated for their efforts.

Here are certain observations which might be of help in revising the manuscript:

1. The authors must define what does active and restrictive approach mean. Throughout the manuscript there is no mention of these two strategies.

2. The authors must describe the active and control group strategies in table 1 and table 2. Without clear description one will not know whether they are comparable.

3. Inclusion criteria: Population is not well defined. The population definition should contain preterm infants with isolated hypotension. This isolated hypotension must be elaborated well like by what definition the diagnosis should be made. As there will be huge variation in the way the condition is defined, and it will have direct impact on the results.

4. Comparator also needs clarity. Which gain takes us back to defining the isolated hypotension.

5. Literature search: The search was limited to only 2 databases. As the authors said that this topic is very controversial with limited literature, it becomes more pertinent to include at least 3 databases. Also, it will be better if the search is updated till end of the year.

6. The authors stated that the definition of IH varied. The variation is so much that it might not be wise to pool them. Particularly if some of the studies did not use BP as a measure to define IH.

7. Outcomes: The authors should provide the risk estimates in text also particularly for important outcomes rather than just referring to Tables and figures.

8. At all places , the number of studies with number of participants must be mentioned along with effect estimates. This provides a wholesome view to the reader. Many of the outcomes are from a handful of participants.

9. Dempsey 2021 compared fluid+ inotrope vs fluid only. How come the first is active approach and second is restrictive

10. Pereira 2019 USA 3-armed pilot RCT: In this study all received fluid bolus and inotropes though at different cut-offs. So technically it should not be combined with Dempsey 2021.

11. The active group of Batton 2016 has significant overlap with Dempsey 2021.

12. How did the authors assess inconsistency? I hope it was limited to statistical heterogeneity.

13. The GRADE profile does not seem to be consistent with recent GRADE recommendations. At many places , especially for imprecision the down gradation should be by 2-3 points. Particularly for RCTs where the cumulative sample size is < 400 and CI are extremely wide.

14. Was Pereira 2019 really at low risk of bias in all domains. It was retrospectively registered with no protocol available at website. Intervention is open label, and many outcomes are subject to observer bias.

15. Many observational studies do not actually meet the criteria for inclusion. They are hugely different from each other with significant overlap between two groups among themselves and with other studies.

Based on above observations the authors should rethink whether the meta-analysis should be done.

6. PLOS authors have the option to publish the peer review history of their article (what does this mean? ). If published, this will include your full peer review and any attached files.

**Do you want your identity to be public for this peer review?** For information about this choice, including consent withdrawal, please see our Privacy Policy .

Reviewer #1: **Yes: ** Srinivas Murki

Reviewer #2: No

---

## [Author Response · Author response to Decision Letter 1]

13 Apr 2024

Response to Editorial Board and Reviewer comments:

Response:

We have formatted the manuscript as per PLOS ONE’s requirements.

2. Please amend either the title on the online submission form (via Edit Submission) or the title in the manuscript so that they are identical.

Response:

We have addressed this aspect.

3. Please amend the manuscript submission data (via Edit Submission) to include author Abdul Kareem Pullattayil S.

Response:

We have amended the manuscript submission data.

4. One of the noted authors is a group [National Neonatal Forum (NNF), Clinical Practice Guidelines (CPG) Group, India]. In addition to naming the author group, please list the individual authors and affiliations within this group in the acknowledgments section of your manuscript. Please also indicate clearly a lead author for this group along with a contact email address.

Response:

We made the suggested changes. (Page 13 -14, Line 285 – 308)

5. Please include your tables as part of your main manuscript and remove the individual files. Please note that supplementary tables (should remain/ be uploaded) as separate ""supporting information"" files".

Response:

We have embedded the three tables within the manuscript. The supplementary file has been uploaded as ‘supporting information file’.

Response:

We have made the suggested changes (Page 20).

Reviewers' comments:

Reviewer's Responses to Questions

Comments to the Author:

1. Is the manuscript technically sound, and do the data support the conclusions?

Reviewer #1: Yes

Reviewer #2: No

2. Has the statistical analysis been performed appropriately and rigorously?

Reviewer #1: Yes

Reviewer #2: No

3. Have the authors made all data underlying the findings in their manuscript fully available?

Reviewer #1: Yes

Reviewer #2: Yes

4. Is the manuscript presented in an intelligible fashion and written in standard English?

Reviewer #1: Yes

Reviewer #2: Yes

Response to the Reviewers:

We thank the Reviewers for their critical appraisal of our manuscript. We have modified the manuscript in accordance with their comments and suggestions.

5. Review Comments to the Author

Reviewer #1: This is good review on the topic Active versus Restrictive management of Isolated Hypotension in preterm infants. Although the evidence evaluated did not result in strong recommendations, the authors suggest the need for well conducted RCT on this important topic.

Response to the Reviewer #1:

We thank the Reviewer for the comments.

I feel the paragraphs on the narrative review may be deleted altogether to improve the readability of the study. However some sections of the narrative review may be used by the authors in the discussion to understand how some the weak or expert recommendations were arrived at.

Response to the Reviewer:

We thank the Reviewer for these suggestions.

Accordingly, we have moved the ‘Narrative review’ section to ‘S1_File’ and labelled the section as ‘Appendix’.

We would also like to bring to the esteemed Reviewers’ attention that we have provided the complete evidence to decision (EtD) framework as per the GRADE working group guidelines in ‘S4 Table’ in ‘S1_File’. Since these recommendations were based on both quantitative meta-analyses and descriptive review of studies that could not be included in the data synthesis, we decided to provide whole EtD framework along with the justifications for providing weak recommendations. We have also indicated this in the manuscript in the discussion section as suggested by the Reviewer. The readership could go in detail the process that was involved in the framing of these recommendations in the EtD framework as indicated in the manuscript. (Page 10, Lines 209-214)

Response to the Reviewer #2:

Reviewer #2: Active Versus Restrictive Approach to Isolated Hypotension In Preterm Neonates: A Systematic Review And Meta-analysis

The authors tried to address an important yet les studied and controversial topic. They have systematically approached the topic and must be congratulated for their efforts.

Response to the Reviewer:

We thank the Reviewer for the encouraging comments.

Here are certain observations which might be of help in revising the manuscript:

1. The authors must define what does active and restrictive approach mean. Throughout the manuscript there is no mention of these two strategies.

Response to the Reviewer:

We thank the Reviewer for this suggestion. We had defined both the approaches in the ‘inclusion criteria’ sub-section (Page 4, Lines 83-100). As you are well aware hypotension was defined varyingly by the authors of the included studies. The following definitions have been commonly used for hypotension in VPT infants: Mean arterial blood pressure (MAP) less than the gestational age, MAP based on centiles, MAP < 30 mm Hg etc. Hence, for such a less studied and debatable topic, we decided to define hypotension based on the primary authors’ / investigators’ definition. We have amended the inclusion criteria by defining isolated hypotension (response to the Reviewer query 3) and the clinical and / or biochemical features of hypoperfusion (the clinical / biochemical criteria for defining hypoperfusion was already mentioned in S4 Table under the sub-section ‘implementation considerations’). We amended the definitions to bring in more clarity in the modified manuscript:

Intervention(I) (active treatment group): Treatment with inotropes for IH. Volume expansion with crystalloids or colloids could have been used prior to inotrope or vasopressor initiation. (Page 4, Lines 89-91)

Comparator(C) (restrictive treatment group):

a. Treatment with volume expansion and / or inotropes in preterm neonates with hypotension only when clinical and / or biochemical features of poor perfusion were present. The clinical and / or biochemical signs of hypoperfusion were defined as presence of either of these: Unexplained tachycardia (>160-170 beats/min), prolonged capillary refilling time (> 3-4 seconds), low peripheral pulses, decreased urine output (< 1 ml/kg/ hour for 4 -6 hours, physiological oliguria or anuria should also be considered), increasing lactate levels (> 3-4 mmol/L) and base deficit (> 8 meq/L).

b. No treatment of IH. (Page 4, Lines 92 – 100)

2. The authors must describe the active and control group strategies in table 1 and table 2. Without clear description one will not know whether they are comparable.

Response to the Reviewer:

We thank the Reviewer for the valuable suggestions. We have included the two approaches (as modified according to the previous suggestion by the Reviewer) in the footnotes in Table 1 and Table 2. (Pages 26 and 38)

3. Inclusion criteria: Population is not well defined. The population definition should contain preterm infants with isolated hypotension. This isolated hypotension must be elaborated well like by what definition the diagnosis should be made. As there will be huge variation in the way the condition is defined, and it will have direct impact on the results.

Response to the Reviewer:

We thank the Reviewer for the suggestions. We have added ‘isolated hypotension’ in the population. The definition of isolated hypotension has also been mentioned. We had a priori defined in our protocol that the MAP cut-off for IH would be ‘as defined by the authors’. The following changes have been made:

“Population(P): Preterm neonates of less than 37 weeks’ gestational age within the first week of life with IH. IH was defined as low mean arterial blood pressure (MAP) as ascertained by the investigator which could be based on definitions such as MAP less than gestational age, MAP value below the 5th centile for the particular gestational age, MAP less than 30 mm Hg without any clinical or biochemical evidence of hypoperfusion.”(Page 4, Lines 84 - 88)

4. Comparator also needs clarity. Which gain takes us back to defining the isolated hypotension.

Response to the Reviewer:

We thank the Reviewer for the comment. This section has been revised as per Reviewer comment 1. (Page 4, Lines 92 – 100)

5. Literature search: The search was limited to only 2 databases. As the authors said that this topic is very controversial with limited literature, it becomes more pertinent to include at least 3 databases. Also, it will be better if the search is updated till end of the year.

Response to the Reviewer:

We thank the Reviewer for the suggestions. We have updated the literature search of the data bases Medline and Embase until 1st April 2024. We have also included another database Web of Science and done a systematic literature search of WOS until 1st April 2024 as well. Accordingly, changes have been made to the PRISMA flow (Figure 1), abstract section (Page 2), literature search strategy section (page 5, line 114), results section (Page 6, Lines 140-146), S1_File: S1b Table and the narrative review of the studies (S1 File: Narrative review). We could include 7 recently published additional studies (References: 46 – 52).

6. The authors stated that the definition of IH varied. The variation is so much that it might not be wise to pool them. Particularly if some of the studies did not use BP as a measure to define IH.

Response to the Reviewer:

We thank the Reviewer for the suggestion. Most studies have used BP as a measure of isolated hypotension. The two RCTs had taken almost similar (if not exactly the same definitions). The results of our meta-analyses also did not reveal any statistical heterogeneity. Hence, we feel that meta-analyses of these two small sized RCTs may possibly be justified.

Coming to observational studies, Faust et al., Aladangady et al., Durrmeyer et al., Kuint et al., Gogcu et al. have all used quite similar definitions to define hypotension which was MAP < GA. Only Dammann et al. had used a different definition (quartile based). We included Dammann et al. study as we presumed the definition used would be quite similar to that of Faust et al. as the BP centiles in Dammann et al. were equivalent to MAP < GA. Batton et al. had used a divergent definition for IH and we did not include the study in the meta-analysis. We were quite conscious of this important aspect pointed out by the Reviewer before proceeding ahead with the meta-analyses. Also, we had addressed this issue by downrating the evidence for indirectness related to definition of IH (population) wherever appropriate (Table 3, explanation ‘b’)(page 42)

7. Outcomes: The authors should provide the risk estimates in text also particularly for important outcomes rather than just referring to Tables and figures.

8. At all places , the number of studies with number of participants must be mentioned along with effect estimates. This provides a wholesome view to the reader. Many of the outcomes are from a handful of participants.

Response to the Reviewer query 7 and 8:

We thank the Reviewer for these suggestions. We have given the effect estimates with 95% CI for all the outcomes along with the number of studies and the sample size. (Pages 8 – 10)

9. Dempsey 2021 compared fluid+ inotrope vs fluid only. How come the first is active approach and second is restrictive

10. Pereira 2019 USA 3-armed pilot RCT: In this study all received fluid bolus and inotropes though at different cut-offs. So technically it should not be combined with Dempsey 2021.

Response to the Reviewer query 9 and 10:

We thank the Reviewer for the query. We would like to state that boluses have been given in many studies including the ones pointed out by the author. As stated in response to Reviewer comment 6, our initial scoping review did bring out these aspects and we decided to do a pragmatic meta-analyses of studies that had quite similar but not exactly the same PICO. As stated before we did not find any significant heterogeneity when pooling the data from these studies. Likewise for the patient population (definition of IH), we had downgraded the evidence for indirectness related to intervention / comparator. As you are well aware, as a Clinical practice Guideline Group we had to make recommendations with the best available evidence that could be generalizable to different settings. Henceforth, we presumed that a pragmatic rather than an explanatory meta-analyses would possibly be the appropriate way forward. The evidence certainty from the meta-analyses of these two RCTs is ‘very low’ now. We have updated them in the manuscript as well. We have included these in the limitations section:

“There were several limitations to this meta-analysis. The definition of isolated hypotension and clinical hypotension with signs of hypoperfusion varied between studies. Further, some of the neonates in the restrictive approach group received volume expansion which could have altered the effect estimates of the meta-analyses. We tried to address these by downrating the evidence by one level for the domain of indirectness related to patient population and the intervention / comparator.” (Page 12, Lines 269 - 275)

We took the decision to do a meta-analysis with a pragmatic approach based on the suggestion by the GRADE group:

“First, if results prove similar from study to study, the variability in study PICOs enhances the generalizability of those results. If patient populations differ widely and results are nevertheless similar across studies, it tells us that patients with different characteristics respond similarly to the intervention, allowing application of the results to a wide range of patients. If studies use different doses but results are nevertheless similar across studies, this tells us intervention effects are independent of dose. Clinicians can then choose freely from among the doses used in the studies (e.g., use lower doses that are less likely to cause adverse effects). Similar interpretations would apply to differences in the comparator and outcomes.”

Reference: https://pubmed.ncbi.nlm.nih.gov/36898507/

In our study, there were differences in “PIC” between the studies, the meta-analyses of most of the outcomes revealed almost similar results. Wherever, significant heterogeneity was dete

---

## [Decision Letter · Decision Letter 1]

30 May 2024

PONE-D-23-44090R1Active Versus Restrictive Approach To Isolated Hypotension In Preterm Neonates: A Systematic Review, Meta-analysis And GRADE based Clinical Practice GuidelinePLOS ONE

Dear Dr. More,

Thank you for submitting your manuscript to PLOS ONE. After careful consideration, we feel that it has merit but does not fully meet PLOS ONE’s publication criteria as it currently stands. Therefore, we invite you to submit a revised version of the manuscript that addresses the points raised during the review process.

We look forward to receiving your revised manuscript.

Kind regards,

Sanjoy Kumer Dey, M.D

Academic Editor

PLOS ONE

Reviewers' comments:

Reviewer's Responses to Questions

**Comments to the Author**

1. If the authors have adequately addressed your comments raised in a previous round of review and you feel that this manuscript is now acceptable for publication, you may indicate that here to bypass the “Comments to the Author” section, enter your conflict of interest statement in the “Confidential to Editor” section, and submit your "Accept" recommendation.

Reviewer #1: All comments have been addressed

Reviewer #2: (No Response)

2. Is the manuscript technically sound, and do the data support the conclusions?

Reviewer #1: Yes

Reviewer #2: Partly

3. Has the statistical analysis been performed appropriately and rigorously? 

Reviewer #1: Yes

Reviewer #2: No

4. Have the authors made all data underlying the findings in their manuscript fully available?

Reviewer #1: Yes

Reviewer #2: Yes

5. Is the manuscript presented in an intelligible fashion and written in standard English?

Reviewer #1: Yes

Reviewer #2: Yes

6. Review Comments to the Author

Reviewer #1: the authors have made all the changes as suggested. The narartive review included in the appendix. Readability improed. No further changes are recommended

Reviewer #2: Thank you considering my comments and revising the manuscript. However i still believe that the studies are not similar to combine them statistically.

7. PLOS authors have the option to publish the peer review history of their article (what does this mean? ). If published, this will include your full peer review and any attached files.

**Do you want your identity to be public for this peer review?** For information about this choice, including consent withdrawal, please see our Privacy Policy .

Reviewer #1: **Yes: ** SRINIVAS MURKI

Reviewer #2: No

---

## [Author Response · Author response to Decision Letter 2]

1 Jun 2024

Dear Dr. More,

Thank you for submitting your manuscript to PLOS ONE. After careful consideration, we feel that it has merit but does not fully meet PLOS ONE’s publication criteria as it currently stands. Therefore, we invite you to submit a revised version of the manuscript that addresses the points raised during the review process.

We look forward to receiving your revised manuscript.

Kind regards,

Sanjoy Kumer Dey, M.D

Academic Editor

PLOS ONE

Response to the Editor:

Dear Sanjoy K Dey, M.D

Academic Editor

PLOS ONE

We would like to thank you for giving us the opportunity to address the Reviewer’s comments and suggestions. We also would like to thank the PLOS ONE team for ensuring a timely re-review of our manuscript.

Sincerely Yours

Viraraghavan V Ramaswamy

On behalf of the Author Team and National Neonatology Forum, India CPG 2023 group

Reviewers' comments:

Reviewer's Responses to Questions

Comments to the Author

1. If the authors have adequately addressed your comments raised in a previous round of review and you feel that this manuscript is now acceptable for publication, you may indicate that here to bypass the “Comments to the Author” section, enter your conflict of interest statement in the “Confidential to Editor” section, and submit your "Accept" recommendation.

Reviewer #1: All comments have been addressed

Reviewer #2: (No Response)

Response to Reviewer #1: We thank the Reviewer for the comment.

2. Is the manuscript technically sound, and do the data support the conclusions?

Reviewer #1: Yes

Reviewer #2: Partly

Response to Reviewer #1 and Reviewer #2: We thank the Reviewers for accepting most of the suggested changes that we had made in the modified manuscript and for the point by point response given to the Reviewer comments.

3. Has the statistical analysis been performed appropriately and rigorously?

Reviewer #1: Yes

Reviewer #2: No

Response to Reviewer #1 and Reviewer #2: We thank the Reviewers for their comments We would like to elaborate with regards to this question to Reviewer #2 comment 6 in the forthcoming section.

4. Have the authors made all data underlying the findings in their manuscript fully available?

Reviewer #1: Yes

Reviewer #2: Yes

Response to Reviewer #1 and Reviewer #2: We thank the Reviewers for the comments.

5. Is the manuscript presented in an intelligible fashion and written in standard English?

Reviewer #1: Yes

Reviewer #2: Yes

Response to Reviewer #1 and Reviewer #2: We thank the Reviewers for the comments.

6. Review Comments to the Author

Reviewer #1: the authors have made all the changes as suggested. The narartive review included in the appendix. Readability improed. No further changes are recommended

Response to Reviewer #1: We thank Reviewer #1 for the suggestions which had greatly improved the quality of the manuscript.

Reviewer #2: Thank you considering my comments and revising the manuscript. However i still believe that the studies are not similar to combine them statistically.

Response to Reviewer #2: We thank Reviewer #2 for all the suggestions and critical appraisal of our initial manuscript. We had extensively revised our manuscript based on the critical inputs given by the Reviewer which made our literature search more comprehensive, the GRADE assessment rigorous, the definition of PICO clearer and finally, we could bring it out with more clarity to the readership that this was a pragmatic systematic review and not an explanatory one. We have detailed these aspects including why we synthesized the data with references from the GRADE working group and what are the implications of that in the revised manuscript.

Page 10-11, Lines 219- 238: “It should be noted that there was some clinical heterogeneity with regards to the patient population, the intervention and the comparator in the included studies in the meta-analyses for the various outcomes. The question of whether pooling of data in a meta-analysis arises in such scenarios. The GRADE working group provides clarity with respect to that. While GRADE visualizes the variability between studies as a source of potential opportunity. GRADE group points out that in some scenarios such variability might even turn out to be one of the strengths of a systematic review.(57) In our systematic review, the meta-analyses of RCTs did not show any between studies heterogeneity though there were some differences in the patient population, intervention and the comparator. This implies that the results of the meta-analysis of the RCTs could be generalizable to different clinical settings where isolated hypotension is defined based various definitions, the intervention and the comparator could deviate slightly still yielding similar clinical outcomes. Further, the meta-analyses of non-RCTs indicated significant heterogeneity for the critical outcome of mortality and the important outcome of severe grade IVH. This provided us an opportunity to explore the various potential reasons for the same. We found postnatal age to be an important effect modifier and hence we could bring out recommendations based on postnatal age of the neonate (≤24 h vs. > 24 h). Such pragmatic systematic reviews form the basis for formulation of clinical practice guidelines for difficult questions that often addressed through expert consensus. Further, these systematic reviews could be generalizable across different settings and also provide clinicians, parents and other stakeholders to choose between different treatment choices.”

---

## [Decision Letter · Decision Letter 2]

31 Jul 2024

PONE-D-23-44090R2Active Versus Restrictive Approach To Isolated Hypotension In Preterm Neonates: A Systematic Review, Meta-analysis And GRADE based Clinical Practice GuidelinePLOS ONE

Dear Dr. More,

Thank you for submitting your manuscript to PLOS ONE. After careful consideration, we feel that it has merit but does not fully meet PLOS ONE’s publication criteria as it currently stands. Therefore, we invite you to submit a revised version of the manuscript that addresses the points raised during the review process.

We look forward to receiving your revised manuscript.

Kind regards,

Sanjoy Kumer Dey, M.D

Academic Editor

PLOS ONE

Reviewers' comments:

Reviewer's Responses to Questions

**Comments to the Author**

1. If the authors have adequately addressed your comments raised in a previous round of review and you feel that this manuscript is now acceptable for publication, you may indicate that here to bypass the “Comments to the Author” section, enter your conflict of interest statement in the “Confidential to Editor” section, and submit your "Accept" recommendation.

Reviewer #2: (No Response)

2. Is the manuscript technically sound, and do the data support the conclusions?

Reviewer #2: Yes

3. Has the statistical analysis been performed appropriately and rigorously? 

Reviewer #2: No

4. Have the authors made all data underlying the findings in their manuscript fully available?

Reviewer #2: Yes

5. Is the manuscript presented in an intelligible fashion and written in standard English?

Reviewer #2: Yes

6. Review Comments to the Author

Reviewer #2: Thank you for providing opportunity to review this manuscript. I have few observations

1. The definitions for intervention and control are hugely different across studies, hence might not be appropriate to combine them.

The following definitions have been commonly used for hypotension in VPT infants: Mean arterial

blood pressure (MAP) less than the gestational age, MAP based on centiles, MAP < 30mm Hg etc. Hence, for such a less studied and debatable topic, we decided to define hypotension based on the primary authors’ / investigators’ definition. We have amended the inclusion criteria by defining isolated hypotension and the clinical and / or biochemical features of hypoperfusion (the clinical / biochemical criteria for defining hypoperfusion was already mentioned in S4

Table under the sub-section ‘implementation considerations’).

Intervention(I) (active treatment group): Treatment with inotropes for IH. Volume

expansion with crystalloids or colloids could have been used prior to inotrope or

vasopressor initiation.

Comparator(C) (restrictive treatment group):

Treatment with volume expansion and/or inotropes in preterm neonates with

hypotension only when clinical and / or biochemical features of poor perfusion were

present. The clinical and / or biochemical signs of hypoperfusion were defined as

presence of either of these: Unexplained tachycardia (>160-170 beats/min), prolonged

capillary refilling time (> 3-4 seconds), low peripheral pulses, decreased urine output (<

1 ml/kg/ hour for 4 -6 hours, physiological oliguria or anuria should also be considered),

increasing lactate levels (> 3-4 mmol/L) and base deficit (> 8 meq/L).

b. No treatment of IH. (Page 4, Lines 92 – 100)

As you can see that the infant who are getting fluid bolus might be considered in intervention by some whereas control in another study.

Two arms of study by Pereiera have been combined together as intervention ( active and moderate) when the two are very different. Also by this way they have done "double counting” of the control group which is not appropriate. That total sample size of two RCts is 118 whereas the authors mentioned it as 139 (Table 3 GRADE). It happened at multiple places because of double counting error.

2. Many observational studies do not actually meet the criteria for inclusion. They are

hugely different from each other with significant overlap between two groups among

themselves and with other studies.

3. I think the manuscript definitely have merits but needs a different way of presentation. The statistical analysis seems incorrect to me. It is like combining apples and oranges and then justifying by looking at statistical heterogeneity (while ignoring clinical heterogeneity)

I am getting a feeling that a narrative review without combing the clinically different studies might be more useful. We agree that there are are extremely limited studies on this aspect and but e they are not alike to be combined together

7. PLOS authors have the option to publish the peer review history of their article (what does this mean? ). If published, this will include your full peer review and any attached files.

**Do you want your identity to be public for this peer review?** For information about this choice, including consent withdrawal, please see our Privacy Policy .

Reviewer #2: No

---

## [Author Response · Author response to Decision Letter 3]

4 Aug 2024

Dear Sanjoy K Dey, M.D 5th August 2024

Academic Editor

PLOS ONE

I thank you for giving us the opportunity to address the Reviewer’s comments and suggestions for the third revision of the manuscript. We have extensively addressed the queries raised by the previous Reviewers backed by published scientific literature on systematic reviews and clinical practice guidelines. Our group have would tried our best to address them in this third revision as well.

Sincerely Yours

Viraraghavan Vadakkencherry Ramaswamy

MD, DM (Neonatology), DNB (Neonatology) (LHMC, Delhi University, Delhi)

Neonatal trainee (OUH, Oslo, Norway)

Clinical Fellow (Oxford University Hospitals, U.K.)

Consultant Neonatologist and Head (Ankura Hospital for Women and Children, KPHB, Hyderabad, India) (2020- )

Associate Task Force Member, Neonatal Life Support, ILCOR (2021-)

Member, Cochrane Neonatal (2021-)

Guest faculty, KDIGO (2023)

Editorial Board Member, BMC Pregnancy and Childbirth (2023-)

Editorial Coordinator, National Neonatal Forum, India Clinical Practice Guidelines Group (2020-)

Editorial Board Member, Journal of Neonatology (2024-)

Member, Global Neonatal Resuscitation Alliance (2024-)

On behalf of the National Neonatology Forum, India Clinical Practice Guidelines Group 2023

Response to the Reviewer

Reviewer #2: Thank you for providing opportunity to review this manuscript. I have few observations

1. The definitions for intervention and control are hugely different across studies, hence might not be appropriate to combine them.

The following definitions have been commonly used for hypotension in VPT infants: Mean arterial blood pressure (MAP) less than the gestational age, MAP based on centiles, MAP < 30mm Hg etc. Hence, for such a less studied and debatable topic, we decided to define hypotension based on the primary authors’ / investigators’ definition. We have amended the inclusion criteria by defining isolated hypotension and the clinical and / or biochemical features of hypoperfusion (the clinical / biochemical criteria for defining hypoperfusion was already mentioned in S4Table under the sub-section ‘implementation considerations’).

Intervention(I) (active treatment group): Treatment with inotropes for IH. Volume

expansion with crystalloids or colloids could have been used prior to inotrope or

vasopressor initiation.

Comparator(C) (restrictive treatment group): Treatment with volume expansion and/or inotropes in preterm neonates with hypotension only when clinical and / or biochemical features of poor perfusion were present. The clinical and / or biochemical signs of hypoperfusion were defined as presence of either of these: Unexplained tachycardia (>160-170 beats/min), prolonged capillary refilling time (> 3-4 seconds), low peripheral pulses, decreased urine output (< 1 ml/kg/ hour for 4 -6 hours, physiological oliguria or anuria should also be considered), increasing lactate levels (> 3-4 mmol/L) and base deficit (> 8 meq/L).

b. No treatment of IH. (Page 4, Lines 92 – 100)

As you can see that the infant who are getting fluid bolus might be considered in intervention by some whereas control in another study.

Two arms of study by Pereiera have been combined together as intervention ( active and moderate) when the two are very different. Also by this way they have done "double counting” of the control group which is not appropriate. That total sample size of two RCts is 118 whereas the authors mentioned it as 139 (Table 3 GRADE). It happened at multiple places because of double counting error.

Response:

1. We would like to thank the Reviewer for the query. It is to be noted that volume expansion in hypotension is a standard of care in the setting of hypotension in most of NICUs across the globe.

A. The HIP trial protocol states “Infants received an infusion of 10mL/kg of 0.9% saline administered over 20 min before commencing their allocated study drug contained in 50 mL syringes identical in appearance.”

B. Pereira et al. trial states “ A written protocol tailored for each gestation and study arm was at the cot side to guide staff, with BP intervention levels. Apart from this target, the guideline (Appendix A) was similar to standard NICU policy. Key elements included triggers for evaluation (BP, tachycardia, impaired perfusion), basic stabilisation, and a maximum of 10-20 mls/kg intravascular volume.” Henceforth, these two trials had used volume expansion in the intervention and control arms prior to randomisation. Coming to observational studies that were included in the meta-analyses most have not stated whether a bolus had been given or not. So, we are unable to comment on this aspect.

2. We would like to thank the Reviewer for pointing out the aspect of ‘double counting’ related to the Pereira et al. RCT in the meta-analyses of the various outcomes. Of the various approaches to avoid the ‘unit-of-analysis’ error that could happen due to this, we decided to adopt the ‘combining groups’ approach suggested by the Cochrane group ( https://onlinelibrary.wiley.com/doi/full/10.1002/cesm.12033). All the meta-analyses for the various outcomes have been re-done. Figure 2 has been modified. The effect estimates and sample sizes of the various outcomes have been changed in the manuscript. Table 3, the summary of findings table has been changed in accordance with the results of the meta-analyses. All these changes have been marked in the marked copy of the revised manuscript.

3. Many observational studies do not actually meet the criteria for inclusion. They are

hugely different from each other with significant overlap between two groups among

themselves and with other studies.

Response:

We thank the Reviewer for the comments. We would like to indicate that this was a systematic review based on which a clinical practice guideline was formulated. Likewise for any clinical practice guideline, we had to adopt a broader inclusion criteria. The specific purpose for the same are two reasons. Firstly, many of these studies would be included in the additional evidence section of the evidence to decision table of the GRADE. I would like to refer to two ILCOR Neonatal Life Support Systematic Reviews

( 1. Ramaswamy VV, Dawson JA, de Almeida MF, Trevisanuto D, Nakwa FL, Kamlin COF, Trang J, Wyckoff MH, Weiner GM, Liley HG; International Liaison Committee on Resuscitation Neonatal Life Support Task Force. Maintaining normothermia immediately after birth in preterm infants <34 weeks' gestation: A systematic review and meta-analysis. Resuscitation. 2023 Oct;191:109934. doi: 10.1016/j.resuscitation.2023.109934. Epub 2023 Aug 18. PMID: 37597649.;

2. Ramaswamy VV, de Almeida MF, Dawson JA, Trevisanuto D, Nakwa FL, Kamlin CO, Hosono S, Wyckoff MH, Liley HG; International Liaison Committee on Resuscitation Neonatal Life Support Task Force. Maintaining normal temperature immediately after birth in late preterm and term infants: A systematic review and meta-analysis. Resuscitation. 2022 Nov;180:81-98. doi: 10.1016/j.resuscitation.2022.09.014. Epub 2022 Sep 27. PMID: 36174764.) where various papers which had not reported on the pre-specified outcomes where included. For example, one paper had evaluated the comfort of doctors and nurses in the OT at different OT temperatures. This was included as an additional evidence in EtD framework. Many QI studies were included where interventions not specified a priori such as education, training etc were components of the QI. To summarise, systematic reviews done for the purpose of formulating CPGs would usually be pragmatic (as it is in this manuscript) while isolated systematic reviews would be more of an explanatory in nature.

3. I think the manuscript definitely have merits but needs a different way of presentation. The statistical analysis seems incorrect to me. It is like combining apples and oranges and then justifying by looking at statistical heterogeneity (while ignoring clinical heterogeneity)

I am getting a feeling that a narrative review without combing the clinically different studies might be more useful. We agree that there are are extremely limited studies on this aspect and but e they are not alike to be combined together.

Response:

We thank the Reviewer for the comments. We would like to emphasize on the following points apart from the response provided to Reviewer query 4 (regarding adopting a pragmatic broader inclusion criteria for the systematic review)

(reference https://www.sciencedirect.com/science/article/pii/S089543562300046X):

A. The GRADE working group emphasizes on statistical heterogeneity more than clinical heterogeneity which is a norm in pragmatic systematic reviews.

B. The GRADE working group also states the following: “First, if results prove similar from study to study, the variability in study PICOs enhances the generalizability of those results. If patient populations differ widely and results are nevertheless similar across studies, it tells us that patients with different characteristics respond similarly to the intervention, allowing application of the results to a wide range of patients. If studies use different doses but results are nevertheless similar across studies, this tells us intervention effects are independent of dose. Clinicians can then choose freely from among the doses used in the studies (e.g., use lower doses that are less likely to cause adverse effects). Similar interpretations would apply to differences in the comparator and outcomes.” From the perspective of our review, the aforementioned statement may be construed as follows:

- Irrespective of the definition of IH used, the restrictive approach might possibly result in better outcomes.

- These results are generalizable to NICUs where volume expansion alone is used (and not inotropes or vasopressors) for treating IH (restrictive approach).

C. We had already stated this aspect in the limitations section for the sake of transparency as follows:

“There were several limitations to this meta-analysis. The definition of isolated hypotension and clinical hypotension with signs of hypoperfusion varied between studies. Further, some of the neonates in the restrictive approach group received volume expansion which could have altered the effect estimates of the meta-analyses. We tried to address these by downrating the evidence by one level for the domain of indirectness related to patient population and the intervention / comparator.”

D. In the previous review, the Reviewer 1 had specifically asked to delete the narrative review section all together which we moved to Appendix in the supplement. Reviewer 1: “I feel the paragraphs on the narrative review may be deleted altogether to improve the readability of the study. However some sections of the narrative review may be used by the authors in the discussion to understand how some the weak or expert recommendations were arrived at.”

Reviewer 1’s response to our edits: “The authors have made all the changes as suggested. The narrative review is included in the appendix. Readability improved. No further changes are recommended”

E. Since we had addressed this issue of this systematic review being predominantly reported based on the results from meta-analyses of various outcomes, we are of the opinion that it would be unfair to make it a narrative review at this stage as the CPG has been widely adopted in our country as well others.

F. Finally, the definition of isolated hypotension did not vary significantly between the studies included in the meta-analyses. The two RCTs had taken almost similar (if not exactly the same definitions). The results of our meta-analyses also did not reveal any statistical heterogeneity as stated before. Hence, we feel that meta-analyses of these two small sized RCTs may possibly be justified. Coming to the observational studies, Faust et al., Aladangady et al., Durrmeyer et al., Kuint et al., and Gogcu et al. have all used quite similar definitions to define hypotension which was MAP < GA. Only Dammann et al. had used a different definition (quartile based). We included Dammann et al. study as we presumed the definition used was quite similar to that of Faust et al.’s as the BP centiles in Dammann et al.’s study were almost similar to MAP < GA. Batton et al. had used a divergent definition for IH and we did not include the study in the meta-analysis. Also, we had addressed this issue by downrating the evidence for indirectness related to definition of IH (population) wherever we deemed appropriate.

---

## [Editor Report · Decision Letter 3]

15 Aug 2024

Active Versus Restrictive Approach To Isolated Hypotension In Preterm Neonates: A Systematic Review, Meta-analysis And GRADE based Clinical Practice Guideline

PONE-D-23-44090R3

Dear Dr.Kiran More,

We’re pleased to inform you that your manuscript has been judged scientifically suitable for publication and will be formally accepted for publication once it meets all outstanding technical requirements.

Kind regards,

Sanjoy Kumer Dey, M.D

Academic Editor

PLOS ONE
---

## [Editor Report · Acceptance letter]

PONE-D-23-44090R3

PLOS ONE

Dear Dr. More,

I'm pleased to inform you that your manuscript has been deemed suitable for publication in PLOS ONE. Congratulations! Your manuscript is now being handed over to our production team.

Kind regards,

on behalf of

Dr. Sanjoy Kumer Dey

Academic Editor

PLOS ONE